# The Importance of Prisons in Achieving Hepatitis C Elimination: Insights from the Australian Experience

**DOI:** 10.3390/v14030497

**Published:** 2022-02-28

**Authors:** Rebecca J. Winter, Jacinta A. Holmes, Timothy J. Papaluca, Alexander J. Thompson

**Affiliations:** 1Behaviours and Health Risks/Disease Elimination Programs, Burnet Institute, Melbourne 3004, Australia; 2Department of Gastroenterology, St Vincent’s Hospital Melbourne, Melbourne 3065, Australia; jacinta.holmes@svha.org.au (J.A.H.); timothy.papaluca@svha.org.au (T.J.P.); alexander.thompson@svha.org.au (A.J.T.); 3School of Public Health and Preventive Medicine, Monash University, Melbourne 3004, Australia; 4Department of Medicine, University of Melbourne, Melbourne 3010, Australia

**Keywords:** hepatitis C virus, prisons, people who inject drugs, disease elimination

## Abstract

Following the availability of highly effective direct-acting antivirals (DAAs) to treat hepatitis C infection, the uptake of treatment by people living with hepatitis C rose dramatically in high- and middle-income countries but has since declined. To achieve the World Health Organization’s (WHO) 2030 target to eliminate hepatitis C as a public health threat among people who inject drugs, an increase in testing and treatment is required, together with improved coverage of harm reduction interventions. The population that remains to be treated in high- and middle-income countries with high hepatitis C prevalence are among the most socially disadvantaged, including people who inject drugs and are involved in the criminal justice system, a group with disproportionate hepatitis C prevalence, compared with people in the wider community. Imprisonment provides an unrivalled opportunity for screening and treating large numbers of people for hepatitis C, who may not access mainstream health services in the community. Despite some implementation challenges, evidence of the efficacy, acceptability, and cost-effectiveness of in-prison hepatitis treatment programs is increasing worldwide, and evaluations of these programs have demonstrated the capacity for treating people in high numbers. In this Perspective we argue that the scale-up of hepatitis C prevention, testing, and treatment programs in prisons, along with the investigation of new and adapted approaches, is critical to achieving WHO elimination goals in many regions; the Australian experience is highlighted as a case example. We conclude by discussing opportunities to improve access to prevention, testing, and treatment for people in prison and other justice-involved populations, including harnessing the changed practices brought about by the COVID-19 pandemic.

## 1. Introduction

Globally, the hepatitis viruses are a leading cause of morbidity and mortality. The World Health Organization (WHO) estimates that 58 million people are currently living with the hepatitis C virus and 290,000 people died from hepatitis C in 2019 [1]. Around 1.5 million new infections are estimated to occur each year [1]. In high-income countries (HIC), most new hepatitis C infections arise as a result of injecting drugs using unsterile injecting equipment, in contrast to low- and middle-income countries, where the epidemic is driven more by unsterile medical practices and contaminated blood and blood products [2,3].

Since 2013, the availability in HIC of all-oral direct-acting antiviral (DAA) medications, which are simple, well-tolerated, and highly effective, has changed the therapeutic landscape for hepatitis C. In people with minimal liver fibrosis, cure of hepatitis C prevents progression to cirrhosis, liver failure, or hepatocellular carcinoma (HCC); in people with cirrhosis, cure is associated with a significant risk reduction. Furthermore, the simplicity and tolerability of DAAs meant that, for the first time, all people living with hepatitis C could be considered for treatment. These prospects led the WHO to set goals to eliminate hepatitis C, as a public health threat, by 2030, reducing incidence by 80%, mortality by 65%, and treating 80% of people living with hepatitis C, underpinned by the principle of ‘leaving no-one behind’ [4]. In HIC, the initial uptake of DAAs among people with known hepatitis C infection was strong; however, as the pool of known cases has been treated over recent years, declines in treatment uptake have been observed—more recently challenged further by the impact of the coronavirus 2019 (COVID-19) pandemic [5,6,7,8,9]. Globally, inequitable access to free or subsidised hepatitis testing, subsidised DAA treatment, and harm and demand reduction interventions, along with poor policy development, implementation, and resourcing, hinders the attainment of the WHO elimination goals for hepatitis C [10]; 80% of HIC are not on track to achieve hepatitis C elimination by 2030 and poorer nations are unlikely to achieve it [6]. Achieving the WHO elimination goals will require the scale up of harm and demand reduction programs, including needle and syringe programs (NSPs) and opioid agonist treatment (OAT), as well as targeted diagnosis and streamlined treatment programs [11,12,13]. The people who remain to be treated are among the most socially disadvantaged and often not engaged in traditional models of hepatitis C care, including people experiencing homelessness, involved in the criminal justice system, and with active drug dependency [14,15,16]. Identifying and treating people with undiagnosed hepatitis C infection requires coordinated national responses, which includes a careful targeting of resources and innovative program design to reach the most marginalised groups and deliver integrated, decentralised, flexible, streamlined, and low-threshold diagnosis and treatment services [2,14,17,18].

The aim of this article is to highlight how the prison system can provide an opportunity for engagement with a disadvantaged and marginalized population and play a crucial role in achieving WHO elimination targets for hepatitis C.

## 2. Hepatitis C and Incarcerated Populations

Over 50% of people who inject drugs are thought to have been exposed to the hepatitis C virus worldwide [19], making people who inject drugs a key population for the targeting of hepatitis C testing and treatment resources. An estimated 15.6 million people, aged 15–64 years, inject drugs globally [19]. The population prevalence of injecting drug use varies considerably between regions, from 0.12% (95% confidence interval [CI]: 0.06, 0.18) in the Middle East and north Africa to 1.30% (95% CI: 0.71, 2.15) in Eastern Europe [19]. In most countries, the possession and use of drugs for non-medical purposes is criminalized, with exceptions including Portugal, The Netherlands, and Switzerland, for example. In countries where drug use is criminalized, people who inject drugs can be highly visible, highly policed, and incarcerated at high rates, leading to a disproportionate imprisonment rate [20]. Globally, around two-thirds of people who inject drugs have experienced incarceration [19]. Therefore, hepatitis C infection disproportionately affects individuals in correctional settings.

Data from September 2018 estimated that there were over 10.7 million people incarcerated in prisons worldwide [21], not including those in pre-trial detention, and approximately 1.5 million people in prison (15%) are estimated to be living with hepatitis C infection [22]. The prevalence of hepatitis C and other infectious disease among imprisoned people who inject drugs is consistently higher than in the general prison population, with imprisoned people who inject drugs having eight times the prevalence of hepatitis C virus (pooled prevalence ratio = 8.1, 95% CI: 6.4, 10.4) than people in prison who do not inject [23]. A systematic review and meta-analysis of studies, conducted up to and including 2012, estimated that the hepatitis C seroprevalence among prison detainees was 26% (95% CI: 23%, 29%), approximating 2.2 million people [24]. For those with a history of injecting drug use, hepatitis C seroprevalence was estimated to be 67% (95% CI: 58%, 75%) among men and 64% (95% CI: 52%, 77%) among women [24]. Spaulding and Thomas have estimated that as many as one million people with undiagnosed hepatitis C infection could be passing through the US correctional system annually [25].

With limited access to demand and harm reduction strategies, prisons are also a high-risk environment for hepatitis C transmission. While studies providing reliable data on hepatitis C incidence in prisons are limited, pooled data have estimated annual transmission risk of 1.4 per 100 person-years (PY) (95% CI: 0.1, 2.7), among the general population of people in prison, and 16.4 per 100 PY (95% CI: 0.8, 32.1), among people in prison with a history of injecting drug use [24]. While many people reduce or cease drug use while in prison [26,27], incarceration can be a catalyst for high-risk behaviour, including the uptake [27,28] or continuation [29] of drug injection at reduced frequency, but with increased syringe sharing [26]. Moreover, imprisonment often represents merely an interruption in drug use trajectories, with a high proportion of people returning to injecting [30] and high-risk injecting behaviour after release [31]. Recent incarceration is associated with a 62% increase in the risk of hepatitis C acquisition (RR 1.62; 95% CI: 1.28, 2.05), [32] and people released from prison, especially those who use drugs, are reincarcerated at high rates [33,34,35], compounding the risk of onward transmission and reinfection.

These data highlight the potential for prison settings to be a critical public health intervention point to improve individual health outcomes and reduce the risk of hepatitis C transmission, both within prison populations and the communities to which people return after their release. Although the prison setting can be a challenging environment for the provision of healthcare, it does provide an unrivalled opportunity for the diagnosis, treatment, and prevention of viral hepatitis and other blood borne viruses in individuals who have difficulty accessing traditional models of care [36]. Despite the WHO and societal guidelines [37,38,39] recommending that people in prison be screened for hepatitis C, in practice implementation varies widely. Of the 124 countries with viral hepatitis testing and treatment plans in place, only 23% have dedicated programs for individuals within correctional facilities [40]. Although up-to-date data are lacking in many regions, an extremely low uptake of hepatitis C screening has been recorded in prisons, with between 20–34% of USA jurisdictions, Canada, and European countries reporting having hepatitis C screening protocols in place [41,42,43].

## 3. Hepatitis C Services in Prisons: Australian Case Study Highlighting the Critical Role of Prisons in HCV Elimination

In Australia, correctional services, including prisons, are the responsibility of the eight states and territories (excluding immigration detention). The 96 prisons [5] are publicly or privately operated, and the delivery of health services is primarily contracted to external providers. This contrasts to the federally funded primary healthcare system available to non-incarcerated Australian residents. Uniquely, however, the provision of DAAs to treat chronic hepatitis C infection has been federally funded and made universally available, including to people in prisons. This funding framework has been instrumental in enabling the implementation of in-prison hepatitis treatment programs across all states and territories. Only non-drug costs such as staffing, pathology, infrastructure, and security are borne by prison healthcare budgets.

Over 80,000 people are incarcerated in Australian prisons annually [44]. In summary, 92% of people in Australian prisons, on the census date in 2021, were male and the median age was 35.7 years for males and 34.9 years for females. The ratio of Aboriginal to non-Aboriginal people in prison was 15.2 for males and 22.8 for females [44]. In concordance with prison populations worldwide, people in Australian prisons have significantly poorer health, compared with people in the wider community [45]. It is estimated that at least 20% of people in Australian prisons are seropositive for hepatitis C, rising above 50% among people who report a history of injecting drug use [46] and disproportionately affecting Aboriginal people [47]. The incidence of hepatitis C infection in prisons in the state of New South Wales was recently observed at 11.4 per 100 PY, among the overall population, and 6.3 per 100 PY, among the population that was continuously imprisoned [48]. Among the continuously imprisoned population, sharing needles/syringes was independently associated with time to seroconversion [48]. Despite being available in the community, to date, no Australian jurisdiction has implemented a prison-based NSP.

In a pilot evaluation of the first 18 months of a nurse-led, state-wide outreach hepatitis program in all prisons in the state of Victoria, Australia, starting in 2015, 949 people in prison underwent a comprehensive hepatitis assessment, including point-of-care liver fibrosis assessment using transient elastography. Seventy-four percent of eligible people (416/562) were commenced on DAA treatment. Of those treated, 86% had never previously engaged in hepatitis C care, and 68% reported injecting in the month prior to incarceration [49]. The program demonstrated extremely high SVR rates (96%) in a per protocol analysis of the 313 prisoners who were treated through the program and retained in care until the SVR12 timepoint [49]. In contrast, among 75 people with hepatitis C infection, who were released to freedom before they could be commenced on treatment, only 19 (25%) were prescribed DAAs within 6 months of release to freedom, 7 of which occurred following re-incarceration [50]. These individuals were all educated regarding linkage to care in the community, with a detailed discharge summary sent to their preferred primary health care physician or practice to streamline hepatitis C treatment. The results suggest that the relative stability of the prison environment may be an opportune environment for successful hepatitis C care, and that high rates of loss to follow-up continue to occur amongst these individuals in the general community.

Beyond successful screening, diagnosis, and treatment, the scale-up of in-prison hepatitis programs can play a leading role in hepatitis C prevention. The Surveillance and Treatment of Prisoners with hepatitis C (SToP-C) study evaluated hepatitis C treatment as prevention in four prisons in the state of New South Wales, Australia, from 2014 to 2019. Approximately 70% of all people imprisoned in these sites were recruited, a total of 3691 participants. After ascertaining participants hepatitis C serostatus after study enrolment, uninfected (n = 2240) and previously infected (n = 725) participants were followed-up every 3 to 6 months to detect primary infection or reinfection, respectively. Participants who were diagnosed with hepatitis C at study enrollment were assessed for treatment and 349/499 eligible participants commenced treatment. Treatment was initially offered through standard prison healthcare services and then scaled up using DAAs via the StoP-C study. Among the at-risk population with longitudinal follow-up, 31% reported injecting drugs during their current imprisonment. The hepatitis C incidence declined by 48%, from 8.31 to 4.35 per 100 person-years, between pre- and post-treatment scale-up periods [51].

In Australia, prison hepatitis programs are playing an increasingly central role in the effort to eliminate hepatitis C as a public health threat. Between March 2016 and February 2017, 2052 treatments were initiated in Australian prisons (an estimated 6% of all treatments nationally) [52]. In 2020, 3005 hepatitis C treatments were initiated in Australia’s prisons, which was estimated to constitute 37% of all treatments nationally [5]. This increase has occurred in direct contrast to the observed decline in treatment uptake in the general community [7]. These statistics underscore both the readiness of people in prison to have their hepatitis C treated and the increasing importance of prison-based hepatitis services in reaching an at-risk population. While there are jurisdictional differences in both the prevalence of hepatitis C among people in prisons and models of hepatitis care that have been implemented, the services share some common features, which have been important factors in their success. Against a back-drop of free access to DAAs and universal screening policies [52], Australian prison hepatitis services are now primary-care led, utilising nurse and GP-led models of care, with specialist support when required [49,53]. The centrality of primary care-led models has improved the accessibility of assessment and treatment for people in prison, through locally delivered prison health services or via in-reach programs, removing the need for transfer to another site to access specialist care. Face-to-face services are provided, and telehealth is utilised when required. This is particularly important in geographically remote and regional prisons, of which there are a number in Australia. A recent global systematic review and meta-analysis identified that decentralised and integrated hepatitis C care and task-shifting to non-specialist roles resulted in improved rates of linkage to care (94% for fully decentralised care [95% CI: 79, 100] vs. 50% for partial/no decentralisation [95% CI: 29, 71] and DAA treatment uptake (72% [95% CI: 48, 91] vs. 39% [95% CI: 17, 63]) among people in prison [17]. The high throughput of high-risk populations seen in prisons means that scaling up hepatitis C programs in prisons is one area that has the potential to increase the yield of new diagnoses and subsequent treatment initiation, contributing to the reduction of new infections and the national burden of disease.

People incarcerated in Australian prisons are generally supportive for in-prison hepatitis C treatment, but also describe apprehension surrounding stigma, discrimination, reinfection risk, treatment side-effects, and a lack of social support, indicating a need for ongoing education and initiatives to promote cultural change [54]. Much work is yet to be done to design and deliver services that meet the needs of all people in prison. In particular, the barriers to accessing in-prison healthcare may be different or compounded for special populations, such as Aboriginal and Torres Strait Islander (First Nations) people, yet there is only one documented Aboriginal community-controlled holistic and culturally-safe primary healthcare provider in one prison in the Australian Capital Territory [55]. For prisoners with cirrhosis, there is also a need to develop sustainable, reliable programs for long-term liver cancer screening in prison, and linkage to primary care following release to freedom—in people with cirrhosis, cure of hepatitis C reduces the risk of future HCC and complications of portal hypertension, but does not abolish it. Guidelines recommend long-term surveillance for HCC and oesophageal varices post-cure in people with cirrhosis [37,39,56]. Care navigators have been shown to be effective for promoting linkage to primary care, following release from prison [57].

The Australian experience highlights the potential for prison health services to access large numbers of hepatitis C positive individuals and individuals with ongoing high-risk behaviours for hepatitis C acquisition, as well as highlighting the opportunity for engagement with hepatitis C care that incarceration can present.

The achievements of Australia’s prison hepatitis programs have, to date, demonstrated feasibility [49,53], cost-effectiveness [58,59], acceptability to people in prison and correctional and health personnel [54,60], and the reduction of hepatitis C prevalence [51,61]. The data also highlight the extremely high rates of injecting drug use among individuals that intersect with the criminal justice system, high rates of re-incarceration, and importantly low rates of engagement either before or following incarceration in traditional community models of hepatitis C care, highlighting the significant health-care gap these individuals experience and key role that prisons play in achieving the WHO elimination targets. Opportunities exist to further expand these programs and improve the cascade of hepatitis care in Australian prisons, including the development and implementation of coordinated national surveillance and monitoring that reports screening and positive test data across all prisons, as well as a cascade of care data [52]. There are also opportunities to improve harm reduction services to ensure that people in prison have the same access to the same standard of care and harm reduction services available to those in the community.

Incorporating hepatitis C screening and treatment programs in correctional settings, ideally using a universal screening approach, will, we argue, be critical for national hepatitis C elimination strategies. Achieving this through combined treatment and prevention, harm minimisation strategies and improvements to the hepatitis C cascade, such as reflex PCR testing, streamlined referrals, and rapid assessment for treatment (including nurse-led models), strategies to improve continuity of care when moving between prison and the community, and biochemical-based non-invasive evaluation of liver fibrosis), will be necessary for optimal success and to achieve the WHO targets [12,62]. Prisons play a critical role and are a key focus for these efforts.

## 4. Future Directions

In many countries, equitable access to DAA treatment and accessible models of prison-based hepatitis C care are yet to be achieved and remain a priority for action. Well-resourced national strategies and reliable data sources for monitoring are also needed. In addition, there are multiple new frontiers to expand and improve on efforts to scale up hepatitis C prevention, testing, and treatment, especially for the people involved in the criminal justice system (Figure 1 and Figure 2).

### 4.1. Integrating Prevention Strategies

A comprehensive approach to hepatitis C management in prisons must recognise the necessity of prevention, alongside testing and treatment programs, incorporating both demand and harm reduction programs. Currently, there is a relative lack of access to interventions equivalent to those available in the community, which have demonstrated effectiveness at reducing risk behaviours or hepatitis C transmission. For example, despite the demonstrated effectiveness in reducing blood-borne virus transmission in community settings, only 10 countries currently have operational NSPs in prisons [63]. We support further implementation/demonstration programs, to expand the evidence base supporting efficacy and safety of NSPs in prisons. At a minimum, cleaning agents, such as bleach, should be available and accessible to all people in prison. Additionally, the provision of OAT in prisons has been shown to reduce the frequency of injection episodes and needle/syringe sharing among people who inject drugs, but is not widely available or accessible to people in prison in many countries [63]. Even countries that offer both continuation and initiation of OAT during imprisonment may limit the medication range offered, provide inadequate dosage, or be plagued with other accessibility challenges [63]. The search for pharmacotherapies to treat methamphetamine dependence is also a vital aspect of current addictions research. Significant proportions of people in prison report the regular injection of methamphetamine [64], and there are currently no pharmacological interventions considered effective treatments [65].

### 4.2. Strategies to Increase Testing

New frontiers for hepatitis C testing are driven both by technological advances and innovations in service delivery, alongside existing evidence-based approaches. Current policy approaches to hepatitis C testing in correctional settings are risk-based, on-demand, or universal. Risk-based testing involves the screening of people for demographic or behavioural risk characteristics, but is estimated to miss up to 75% of infections [66]. It is also thought to contribute to the reinforcement of stigma through targeting people who use drugs [67]. On-demand testing is performed only on request; this approach relies on individual agency to seek care and assumes a high degree of health literacy, which is known to be poor in prison populations [68,69]. Under a universal testing policy, testing is available to all people in prison, which may be provided on an opt-in or -out basis. Opt-in frameworks offer testing, whereas under an opt-out framework all people in prison are tested unless they decline. Carefully implemented, universal, opt-out testing policies can improve testing uptake and hepatitis C diagnoses and have the benefit of normalising hepatitis C testing, potentially reducing the associated stigma [70,71,72]. Universal opt-out testing policies are endorsed by the American Association for the Study of Liver Diseases [38], European Association for the Study of the Liver [37], and US Federal Bureau of Prisons [73].

Technologies such as point-of-care testing may increase case detection, reduce the steps required for diagnosis, and facilitate rapid pathways to treatment and care, where capacity exists [74]. In the prison setting, the major benefit of the use of point-of-care testing has the potential for increased case detection at prison reception. Additional advantages include the reduced need for multiple appointments and blood collection, allowing for a quick appraisal of serostatus and immediate referral for further assessment or treatment initiation. In resource-poor settings, point-of-care tests can be a viable alternative to pathology. The available hepatitis C point-of-care tests include antigen tests using oral fluid swabs or blood samples (e.g., OraQuick HCV^®^, OraSure Technologies, Bethlehem, PA, USA), which provide a result in 20–40 min, and viral load tests using capillary blood samples (e.g., Xpert^®^ HCV VL Fingerstick assay, Cepheid, Sunnyvale, CA, USA), which provide a result in ~60 min. In a UK trial comparing the use of dried blood spot (DBS) testing versus the OraQuick HCV and Xpert^®^ HCV VL Fingerstick assay for screening and confirmatory testing, the median time from sample collection to treatment initiation was 50 days in the DBS group and 4 days in the point-of-care group [74]. A recent Australian before–after historically controlled study compared hepatitis C testing and treatment uptake using a ‘one stop shop’ model of care utilising the Xpert^®^ HCV VL Fingerstick assay (n = 301) and standard of care (venous blood sample; n = 240) at one reception prison. The uptake of hepatitis C antibody/RNA testing was 99% during the intervention phase, compared with 45% during the standard of care phase, and the proportion of participants initiating DAA treatment within 12 weeks from enrollment was 93% vs. 26%. The median time to treatment initiation was 6 days (IQR: 5–6) and 90 days (IQR: 62–127) during the standard of care phase [75].

### 4.3. Beyond the Prison Walls

While the closed nature of prison settings has the potential to lend some unique advantages to the delivery of hepatitis C services across the care continuum [76], the delivery of in-prison healthcare should not be considered in isolation from community settings. The provision of custodial and community health services is often siloed, and effective referral and supported linkage to care for people entering and leaving prisons is mostly inadequate [77]. To support the bridging of this gap, some commentators have called for the transfer of responsibility for correctional healthcare to local health authorities, rather than corrective services [61].

One of the challenges for the provision of health care for people in prison is the dynamics of movement, both within and between prisons, which can disrupt the commencement or continuation of treatment regimens and care [50,57,76]. In addition, due to a multitude of factors, people leaving prison face unique challenges upon re-entering the community and confront barriers, related to social and structural determinants of health, that make accessing traditional models of healthcare difficult, including hepatitis C care. Barriers include poor social support, housing instability, poverty and financial concerns, meaningful and reliable employment, poor mental and physical health, and the resumption or continuation of drug and/or alcohol use [30,34,57,78]. Furthermore, perceived (and actual) stigma and discrimination, low health literacy, and poor awareness of advancements in hepatitis C treatment may further contribute to the lack of engagement.

Akiyama and colleagues conducted a single-arm trial of transitional care coordination for people released from New York City jails and successfully linked 31% (26/84) to hepatitis C care, within a median of 20.5 days of their release [76]. A pilot study conducted by authors of this commentary in the state of Victoria, Australia randomised people being released from prison with untreated hepatitis C infection, to either supported care navigation or standard care (referral to community-based primary care provider). Of the 46 participants, those receiving the intervention were significantly more likely to commence DAA treatment than those assigned standard care (73%, [16/22] vs. 33% [8/24]), and the median time to treatment initiation was shorter (21 days [IQR: 11–42] vs. 82 days [IQR: 44–99]) among those who accessed DAA treatment within six months [57]. An Iranian study provided integrated hepatitis C care to all people incarcerated in a provincial prison and facilitated a referral to local district health services if participants were released prior to DAA treatment initiation or completion. Participants were provided with five days of DAA medication at release and healthcare providers at district health services were provided with contact details to actively contact and engage released participants [79]. Of the released participants, 68% (30/44) were linked to care in the community and 70% (21/30) completed treatment, including 60% (12/20) and 90% (9/10) among those who were released before and during treatment, respectively [79].

Further, prison is one arm of a multicomponent criminal justice system, comprised also of police detention, the courts, and community-based corrective services. Entry points to hepatitis C testing and care, as well as opportunities to integrate continuity in care, should be considered for justice-involved groups in non-custodial settings. For example, in Australia there were 43,073 people in prison to the June quarter 2021, while the number of people serving supervised correctional orders in the community was 78,785 during the same period [80]. The existing infrastructure of community corrective services, coupled with regular and frequent contact with clients, offers an avenue for accessing a population, which likely share similar hepatitis C risk factors to people imprisoned. There are few documented attempts to access this population published in the academic literature. One study from Spain reported testing 93% (548/590) of people at a single site and 81% of those with detectable RNA commenced treatment; only 7% of the sample had previously known their hepatitis C serological status [81]. A community-led project in Brisbane, Australia offers general, practitioner-led, on-site care at three community corrections offices, but the results are not yet published.

### 4.4. Global COVID-19 Pandemic and HCV Elimination Efforts

Since the first case of COVID-19 was identified in December 2019, the global COVID-19 pandemic has resulted in significant morbidity and mortality. At the time of writing, there have been 276,436,619 confirmed cases and 5,374,744 deaths reported to the WHO worldwide [82]. This has led to significant strain on healthcare systems, with many countries facing disruption to and overwhelming pressure on healthcare services for COVID-19-related care. Furthermore, strict public health measures have been implemented, to varying degrees around the world, to curtail viral transmission. These have included personal and societal social distancing measures, such as density limits, wearing of masks, and lockdowns, as well as modifications to healthcare delivery, such as a move towards remote telemedicine consultations, relocation of services to make space for COVID-19 related activities, redeployment of staff, and cessation of “non-urgent” programs and services. These have led to significant disruption to hepatitis C elimination efforts, due to patient-related factors (such as reluctance to attend or seek in-person medical attention at healthcare providers), laboratory-related factors (such as delays to processing hepatitis C RNA specimens due to overwhelming COVID-19 testing requirements), access-related factors (such as redeployment of key hepatitis C elimination program workforces and limited access to persons living with hepatitis C in some settings, such as the custodial setting), and reduction and, in some cases, complete cessation of education programs for hepatitis C elimination.

The Coalition for Global Hepatitis Elimination recently conducted a survey with hepatitis C program managers and clinicians to assess the impact of COVID-19 on the delivery of hepatitis B and C testing and treatment services, redeployment of staff for COVID-19-related duties, and any perceived benefits to hepatitis services emerging from the COVID-19 response [83]. In addition, supply chain and training interruptions were evaluated. Survey respondents included 103 clinicians and program managers from 44 countries across the six WHO regions, and the majority were physicians or nurses providing care in public hospitals and academic centres (72%). The main findings were that the majority deferred in-person clinic visits, which remained lower than the pre-COVID-19 era, following the peak of COVID-19 infections during the study period, due to patients’ anxiety, fear, and loss of staff and space. Most participants (88%) reported disruption to hepatitis C services, 39% reported a >50% decline in the number of patients being treated for hepatitis C, and 28% reported a >50% decline in hepatitis C testing. Furthermore, 79% reported disruptions to ancillary services (such as OAT or NSPs), and many respondents reported DAA or testing supply chain disruptions. Reported deferral of imaging and laboratory testing was common (63–77%) and most patient contact (>75%) was via telemedicine. With regards to COVID-19 redeployment, 88% reported changes in work allocation, with 31% reporting > 25% of time re-directed solely to COVID-19 care. However, participants also highlighted several aspects of the COVID-19 response that were perceived to potentially be beneficial to hepatitis care, including increased laboratory testing, improved healthcare worker training, contact tracing, surveillance, and referral networks. Similar reductions in hepatitis care and the number of patients receiving DAAs has been observed in other reports [8,9]. In addition, countries that were previously on track to meet the 2030 WHO elimination target are now no longer likely to meet this target, although the full impact is not yet known. In a modelling study, adapted for 110 countries to include a “no delay” status quo and a “1 year delay” scenario to evaluate the incremental change in hepatitis C liver-related deaths and liver cancer, following a hiatus in hepatitis programs, assuming significant disruption to screening, diagnosis, and treatment, and weighted to calculate regional and global estimates [84], a “one year delay” scenario resulted in 44,800 and 72,300 excess liver cancer and liver-related deaths, respectively, relative to the “no delay” scenario.

Regarding prisons, the COVID-19 pandemic has precipitated calls by public health experts for decarceration strategies, due to the high-risk of transmission in closed settings and high prevalence of co-morbidities among people in prison, which make them susceptible to severe COVID disease [85,86]. Additional recommendations in the literature include quarantining of people entering prison before release into the general prison population, visitor restrictions, screening of staff and people imprisoned, and priority vaccination programs, among a host of other measures [86]. While information on the impact of COVID-19 management on in-prison hepatitis C care and other health services is scarce, services are likely to have suffered similar impacts to those in the community. As the pandemic progresses and COVID-19 mitigation and management strategies become ‘business as usual’ in both community and correctional settings, there is an opportunity to review the advantages and shortcomings of the implemented protocols and practices to inform the adaptation of other public health interventions, including hepatitis C programs. For example, quarantine periods may present an opportunity for more comprehensive health needs assessments for people entering prison, including hepatitis C screening and assessment. The expansion of remote delivery of care using telehealth may allow for accessing a greater number of people in prison. Upscaling of COVID-19 viral testing may provide an established and accepted pathway for improving hepatitis C screening practices, and improved training, contact tracing, referral, and surveillance strategies may lead to further innovations and improved models of hepatitis C care delivery.

## 5. Summary

Fundamental to the provision of healthcare in prisons is the principle that all people have the right to healthcare equivalent to that available in the community (the ‘Mandela rule’) [87]. The availability of highly effective DAAs to treat hepatitis C, paired with effective strategies for harm reduction in community settings, and high prevalence of hepatitis C infection among people in prison provides a compelling rationale for their availability in prisons on an equivalent basis. Aside from the individual health benefits gained in treating hepatitis C infection, there are notable public health benefits, including reduced morbidity and mortality. Investment in early case detection and treatment, including those among prison populations, will reap rewards in community health and future healthcare costs. To prevent unnecessary hepatitis C-related complications and deaths, attention should be re-focused, as soon as feasible, on the rejuvenation of hepatitis C elimination programs, notwithstanding the COVID-19 pandemic, which, despite challenges to date, may even create new opportunities to strengthen hepatitis C elimination efforts.

## Figures and Tables

**Figure 1 viruses-14-00497-f001:**
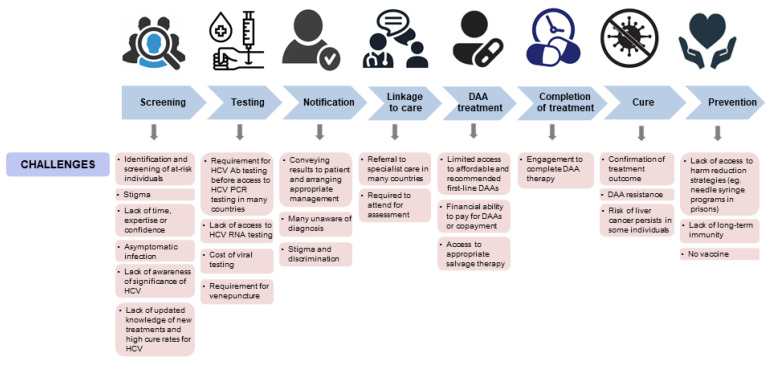
Current challenges in hepatitis C care. DAA = direct acting antivirals; HCV = hepatitis C virus; Ab = antibody; PCR = polymerase chain reaction; RNA = ribonucleic acid.

**Figure 2 viruses-14-00497-f002:**
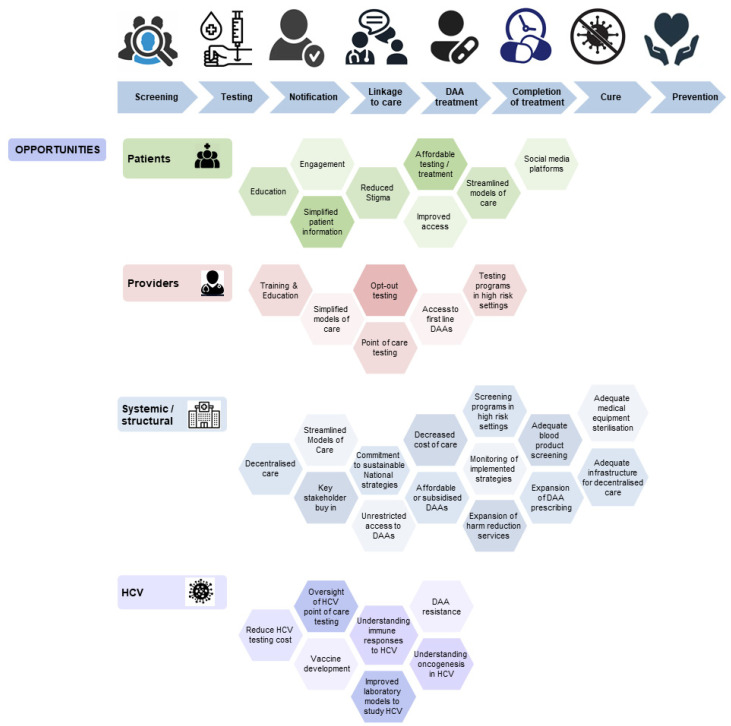
Opportunities in hepatitis C care. DAA = direct acting antivirals; HCV = hepatitis C virus.

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
