# Peer review of "The Importance of Prisons in Achieving Hepatitis C Elimination: Insights from the Australian Experience"

_viruses, 2022, doi:10.3390/v14030497_

Round 1
Reviewer 1 Report
General comments:
- Avoid expression as "there is" in scientific writing.
- It is not clear what the article will present (unclear abstract and introduction).
- It is not clear why the subject is important and for whom is importnat (e.g., I live in a country where the IUD is not frequent).
- It is not clear why the specific case presented is important for scientific community and which is the generalizability of this particular experience.
- I did not catck the perspective.
Abstract:
- Please define disproportionate in "disproportionate hepatitis C prevalence".
- The abstract in uninformative: the problem is appropriately presented but no information regarding the content and originality of the article is presented. It is not clear what it will be presented in the article, why and the inportance of the topic.
Introduction:
- please update the information reported for 2019.
- Please also report the IUD by countries.
- state at the end of this section the aim of the article.
Hepatitis C and incarcerated populations:
- "In most countries the possession and use of drugs for non-medical purposes is criminalised." please list these contries or those in which are not.
- "high rates" please be specific; what high means? it is 80%?
- "Around two-thirds of" where?
- "1·62" should be read as "1.62". Similar for other similar numbers.
- "use drugs – are reincarcerated at high rates" please be specific and provide definition for high.
4. Future directions:
- The quality of fig.1 is very low.
5. Conclusion: I do not see why a perspective must have a conclusion. The conlcusion must be supported by own results and does not fit to perspective articles.
Reviewer 2 Report
Global hepatitis C eradication is a challenging task. Identifying and treating citizens with undiagnosed and non-eradicated hepatitis C require coordinated and prompt global and national efforts. It is interesting to highlight and elucidate that imprisonment has provided an opportunity for screening, treating and following up on large-scale numbers of vulnerable, incarcerated persons with hepatitis C to benefit risk reduction (with kinetics over time on and off treatment) for the entire community. Hepatologists worldwide have already been impressed by similar, large-scale, and real-life experiences, and administrative and hospital datasets as well as presentations the past decade.
Major comment:
= The present article addressing perspectives is an interesting addition to the current understanding of the importance of prisons in achieving national hepatitis C elimination
= It is recommended to make the statements substantial and non-redundant
= As judged by clinical hepatologists… (who have worked on the following concepts through elastography-based surveillance and paired liver histology the past decade: antiviral treatment matters/kinetics across the treatment timeline: early, rapid necro-inflammatory resolution and stiffness decline/late, slow fibrosis and collagen content regression and stiffness decline/the baseline of post-viral eradication surveillance using liver stiffness should be obtained post-viral eradication)…
= It would be more interesting than the present to appropriately, concisely and substantially incorporate into the present article (abstract, introduction, discussions…) 5 to 6 sentences or more by addressing or pointing out in a few words or brief phrases the following concepts, and by citing 2 to 3 published studies indexed by key words—prison, elastography, etc.:
-Concept: The incidence of adverse liver-related events has been reported to be not zero starting from the baseline post-viral eradication. The residual or left-over posttreatment burden of liver fibrosis and portal pressure may constitute the common and pivotal upstream mechanism which translates into the development of liver-related events, related to carcinogenesis and portal hypertension. These adverse events remain emerging problems in the increasing population post-viral eradication (Hepatologists are currently highly concerned about the issues regarding the “posttreatment” follow-ups or surveillance! Year by year only “treated” subjects are seen in developed countries or regions worldwide or in our Asia-Pacific region. Treatment success is no longer a problem.) (Posttreatment issues are scant in the present perspectives!)
-Concept: antiviral treatment matters/ treatments result in declining kinetics in liver stiffness, noninvasive tests, and liver collagen content
-Concept: liver collagen content declines over time/ liver fibrosis burden declines over time/ left-over post-viral eradication liver stiffness through elastography indicates the left-over liver fibrosis burden post-viral eradication in kinetics
-Concept: cutoff values of liver stiffness for liver fibrosis assessment vary and decline over time through treatment/ lower cutoffs are required post-viral eradication (EASL Guidelines 2021)
-Concept: the baseline to start posttreatment prognostics should be obtained post-viral eradication! (Barr Radiology 2020; EASL Guidelines 2021)
-Concept: the kinetics in these tests over time reflect the changing risk stratifications over time/ test results and risks vary over time on and off treatment
-Concept: recall and reminder policy in prisons during posttreatment follow-ups are warranted/ what are actionable and acceptable pre-specified measures to react to a positive or abnormal test that designate prisoners as high-risk during post-viral eradication follow-ups?/ still warranted in real life/ at least address in a few words
Reviewer 3 Report
The authors analyzed the problem of eliminating HCV infection, according to the WHO recommendations for the eradication of HCV infection by 2030. They focused on the difficult population of prisoners with a high proportion of drug users with HCV co-infection. The authors point out that the prison offers beneficial opportunities to diagnose, treat and prevent hepatitis C. Dedicated HCV eradication programs among prisoners could bring the world closer to achieving WHO goals. The authors described the epidemiological situation of hepatitis C in the prisoners' environment very well. They emphasized the possibility of introducing HCV treatment programs with the use of DDA drugs, with high effectiveness. The work is a valuable study of the public health problem of Australia and many countries of the world. The scheme presented in the paper is worth implementing by public health care services in order to more effectively combat and prevent hepatitis C in different populations, also in prisoners. I recommend the paper for publication in the Viruses.
Reviewer 4 Report
Thank you for the opportunity to review the manuscript “The importance of prisons in achieving hepatitis C elimination: insights from the Australian experience”. The Authors note that after an initial increase in the number of antiviral DAA therapies in the general population in high- and middle-income countries, there was a subsequent decline. This makes it necessary to look for subpopulations that require the intensification of activities. Testing and subsequent treatment of HCV-infected prisoners contribute to the microelimination of HCV, facilitating the achievement of the WHO goal. The issue was presented based on the experience of Australia, the place where the world-first project offering opportunities to halt the transmission of HCV in prisons (StoP-C) was implemented. The manuscript is clear, comprehensive, and relevant for the field, and is presented in a well-structured manner. The cited literature is properly selected and comes from recent years. Statements and conclusions are drawn coherently and are supported by the data. In my opinion, the paper is suitable for publication in “Viruses”, however, I have some minor comments. Figure 1 is, in my opinion, not very readable, too much information is contained in a small area. The References are not prepared in accordance with the requirements of the journal.
Round 2
Reviewer 1 Report
The current version of the manuscript is more clear.
Some minor points:
- Since the type of the manuscript is "Perspective" I would recommend to rename the cocnlusions section. Furthemore, references had no place in the section Conclusion(s). This section sound more like "To sum up".
- Reference section do not follow the formatting journal requirements. Some references are incomplete: 1, 2, 4, 5, 21, 39, 43, 44, 45, 46, 64, 78, 80, and 89 as web pages and 17 as an article. I would no recommend citation of a reference which is under review if it is not deposited in a pre-print database.
Author Response
Thank you for your comments and careful review of the manuscript.

Reviewer 2 Report
The present manuscript is well revised.
Author Response
Thank you for your feedback and craeful review of the manuscript.
